# A novel humanized mouse lacking murine P450 oxidoreductase for studying human drug metabolism

Mercedes Barzi[1], Francis P. Pankowicz[1,2], Barry Zorman[3], Xing Liu[4], Xavier Legras[1], Diane Yang[1,2], Malgorzata Borowiak[1,2,5,6,7,8], Beatrice Bissig-Choisat[1,5], Pavel Sumazin[3,7], Feng Li[4,5] & Karl-Dimiter Bissig[1,2,5,6,7]

Only one out of 10 drugs in development passes clinical trials. Many fail because experimental animal models poorly predict human xenobiotic metabolism. Human liver chimeric mice are a step forward in this regard, as the human hepatocytes in chimeric livers generate human metabolites, but the remaining murine hepatocytes contain an expanded set of P450 cytochromes that form the major class of drug-metabolizing enzymes. We therefore generated a conditional knock-out of the NADPH-P450 oxidoreductase (*Por*) gene combined with *Il2rg*$^{-/-}$/*Rag2*$^{-/-}$/*Fah*$^{-/-}$ (PIRF) mice. Here we show that homozygous PIRF mouse livers are readily repopulated with human hepatocytes, and when the murine *Por* gene is deleted (<5%), they predominantly use human cytochrome metabolism. When given the anticancer drug gefitinib or the retroviral drug atazanavir, the *Por*-deleted humanized PIRF mice develop higher levels of the major human metabolites than current models. Humanized, murine Por-deficient PIRF mice can thus predict human drug metabolism and should be useful for preclinical drug development.

[1] Center for Cell and Gene Therapy, Stem Cells and Regenerative Medicine Center, Baylor College of Medicine, Houston TX-77030, USA. [2] Graduate Program, Department of Molecular and Cellular Biology, Baylor College of Medicine, Houston TX-77030, USA. [3] Texas Children's Cancer Center, Department of Pediatrics, Baylor College of Medicine, Houston TX-77030, USA. [4] Alkek Center for Molecular Discovery, Advanced Technology Core, Baylor College of Medicine, Houston TX-77030, USA. [5] Department of Molecular and Cellular Biology, Baylor College of Medicine, Houston TX-77030, USA. [6] Program in Developmental Biology, Baylor College of Medicine, Houston TX-77030, USA. [7] Dan L. Duncan Cancer Center, Baylor College of Medicine, Houston TX-77030, USA. [8] McNair Medical Institute, Houston TX-77030, USA. Correspondence and requests for materials should be addressed to K.-D.B. (email: bissig@bcm.edu)

Hepatotoxicity and cutaneous hypersensitivity reactions are the most common reasons for toxicity-related termination of drugs in human clinical trials[1], and they are also the most poorly predicted by results in animal studies. The difficulty is that drug metabolism depends on NADPH-P450 cytochromes, but whereas human cytochrome clusters contain 27 putatively functional genes, the mouse cytochrome clusters contain nearly three times as many[2]. More to the point, the functions of many murine and human cytochromes have yet to be elucidated, and the interspecies matches of known functions are not particularly good[3]. This makes accurate prediction of human drug metabolism in the mouse quite challenging.

Since the primary site of drug metabolism is the liver, human liver chimeric mice are an attractive alternative to conventional animal models. Isolated human hepatocyte can repopulate the murine liver of several mouse models[4–11] and several groups have used human liver chimeric mice for evaluation of drug toxicity[12–16] and identification of human metabolites[17–19]. The shortcoming of these chimeric mice is the remaining murine liver tissue. We have previously shown that human liver chimerism averages 42% in the $Fah^{-/-}/Rag2^{-/-}/Il2rg^{-/-}$ (FRG) strain and maximal human chimerism (~95%) is reached in <5% of animals[20]. Moreover, high chimerism on the cellular level does not take into account different expression levels and affinities of murine drug metabolizing enzymes. Only a complete liver humanization would allow human drug metabolism, but considering the many interspecies incompatibilities of ligands and receptors this might be a very challenging task. An alternative approach would be to inhibit drug metabolism in the remaining murine liver. Indeed, recently the urokinase-type plasminogen activator/severe combined immunodeficiency (uPA/SCID) mouse[21] have been crossed with the cytochrome 3a (Cyp3a) knock-out mouse, generating a strain which can be repopulated with human hepatocytes and is deficient of one of the main human drug metabolizing cytochrome clusters[22, 23]. Unfortunately, these mice massively upregulate several other cytochrome clusters, which defeats the purpose of the model[22]. A logical next step would be to delete other murine cytochrome clusters in this model, but considering that there are 72 functional genes and probably a similar number of pseudogenes in the murine genome, this would be a major challenge.

Despite the magnitude of the cytochrome P450 family, all cytochromes have only one electron donor, the NADPH-P450 oxidoreductase (Por). Deletion of the Por gene is embryonic-lethal (E10.5) because the enzyme is involved in a variety of essential metabolic pathways[24]. Nevertheless, liver-restricted deletion of Por leads to healthy mice with some metabolic deficiencies in cytochrome, cholesterol, and heme metabolism[25–27].

To develop a better animal model for human-specific drug metabolism, we generated a conditional Por mouse and deleted the Il2rg, Rag2 and Fah genes in the homozygous zygotes. We show that this PIRF strain is readily repopulated with human hepatocytes and reflects human xenobiotic metabolism upon deletion of the murine Por gene. Thus, these PIRF mice could be used to predict human drug metabolism and should improve preclinical drug development.

## Results

### Generation of novel mouse model for hepatocyte repopulation.
In order to functionally block murine cytochrome metabolism, we generated a conditional (floxed exon 3 and 4) knock-out of the NADPH-P450 oxidoreductase (Por) gene by targeting mouse embryonic stem cells (ESCs)[28] (Supplementary Fig. 1). Injected blastocysts with properly targeted ESCs produced chimeras with germline transmission of the Por "knock-out first" allele[29]. We confirmed expression from the targeted Por locus using the lacZ

expression cassette in the embryo and adult liver (Supplementary Fig. 1c). We next bred the mice with a flippase-expressing strain[30] to generate a CRE recombinase conditional Por knock-out strain ($Por^{c/c}$). Homozygous zygotes from this strain were injected with the bacterial type II clustered regularly interspaced short palindromic repeats/Cas9 (CRISPR-Cas9) system[31–33] targeting simultaneous deletion of critical exons of the Il2rg, Rag2, and Fah genes (Supplementary Figs 2 and 3) to generate the PIRF strain (Fig. 1a). Homozygous PIRF mice are thus immune-deficient, lacking T, B, and NK cells, but are healthy and fertile. Since adenoviral gene therapy vectors efficiently transduce hepatocytes in vivo, we deleted the Por gene using an adenovirus coding CRE recombinase (Adeno-CRE). We injected increasing doses (2.2 × $10^{8-10}$ per mouse) of the virus intravenously into PIRF mice. Quantitative RT-PCR of the Por mRNA in liver revealed efficient deletion only at high doses of adenovirus (Fig. 1b). Immunostaining for Por (Fig. 1c) confirmed these findings, although a minimal residual signal could be detected by western blotting even at the highest dose used (Fig. 1d). Por-deleted PIRF mouse livers accumulated lipids starting about 2 weeks after adenoviral transduction (Fig. 2a), but in contrast to the immune competent $Alb-Cre/Por^{c/c}$ strain[25, 27], without infiltration and lacking necrosis (Fig. 2b). Nevertheless, residual Por expressing hepatocytes had a growth advantage over the lipid-rich Por-deleted hepatocytes, and clonal expansion of a few Por expressing cells could be detected 4 weeks after adenoviral transduction by immunostaining (Fig. 2c).

### Characterization of humanized PIRF mice.
We next generated human liver chimeric mice using the PIRF strain[5, 20, 34]. To ensure cytochrome P450 metabolism would be human-specific, we injected Adeno-Cre (2.2 × $10^{10}$ pfu per mouse) before human hepatocyte transplantation and an additional dose of Adeno-Cre in some highly humanized PIRF (Hu-PIRF) mice. Immunostaining revealed that an almost complete deletion of the Por gene could be achieved only in double-injected humanized PIRF (Hu-PIRF 2x) mice (Fig. 3a). Quantitative PCR and western blotting corroborated the massive reduction of murine Por upon adenoviral delivery of CRE (Supplementary Fig. 4). We next performed gene expression profiling to compare PIRF mice repopulated with human hepatocytes (Hu-PIRF) that were injected with either Adeno-CRE (Hu-PIRF 2x) or Adeno-GFP (Fig. 3b). Both groups were repopulated with human hepatocytes from the same hepatocyte donors (Supplementary Table 1) to avoid inter-individual variations.

Expression of the murine P450 cytochromes was clearly altered for 27 out of 38 genes analyzed after Por deletion (Fig. 3c): 24 cytochromes were significantly upregulated (1.5–12.5-fold) and 3 cytochromes significantly downregulated (0.5–0.3-fold). The expression profiles of these murine cytochromes were by in large comparable to those from previous work in non-humanized, Por-deficient mice (Supplementary Table 2)[35]. In the human part of the same chimeric liver, human P450 cytochromes were less altered upon deletion of murine Por (Fig. 3d). Half of the human cytochromes were only slightly altered (0.5–1.5-fold change), while the other half were moderately upregulated (1.5–2.4-fold).

Not all human cytochromes serve an important role in xenobiotic metabolism. From the 200 most-prescribed drugs in the United States, about three-quarter are metabolized through P450 cytochromes, of which CYP3A4/5, 2C9, 2C19, 2D6, and 1A2 contribute to ~95%[36]. We compared these human cytochrome clusters from chimeric livers (Hu-PIRF 2x) with the originating, isogenic primary hepatocytes. For this comparison, we again used two donor hepatocytes (Supplementary Table 1) and the corresponding human (isogenic) liver chimeric

mice ($N = 6$). Expression levels were similar for most clusters, and these important cytochromes were all robustly expressed in chimeric livers (Fig. 3e). Interestingly, some human clusters (CYP1A2, CYP2B6, CYP2C19, and CYP3A4) were expressed at even higher levels in the chimeric liver than in primary human hepatocytes.

**Xenobiotic metabolism of humanized PIRF mice.** To validate Hu-PIRF mice for human drug metabolism, we studied xenobiotic metabolism of gefitinib[37], an inhibitor of epidermal growth factor receptor used against lung cancer and a variety of other neoplasia[38]. Gefitinib is metabolized primarily by the P450 cytochrome system, including CYP3A4 and 2D6. We recently identified new gefitinib metabolites and demonstrated considerable differences between human and mouse liver microsomes[39], but regardless of dose, route, or species, gefitinib is excreted primarily in the feces (<7% in the urine)[40, 41]. We therefore analyzed the feces of non-humanized PIRF mice for gefitinib metabolites during the first 16 h after intravenous injection of gefitinib.

Mass spectrometry revealed a reduction of several gefitinib metabolites upon deletion of the Por gene, implying a Por-dependent P450 cytochrome deficiency for these metabolites (Fig. 4a and Supplementary Fig. 5). Since some metabolites were not significantly altered, we tested the possibility that residual Por activity was responsible for persistent murine P450 cytochrome metabolism in our system. We crossed the $Por^{c/c}$ strain with a transgenic mouse that expresses CRE under the albumin promoter. Por protein in the liver of $Alb\text{-}CRE/Por^{c/c}$ animals was efficiently deleted (Supplementary Fig. 6); nevertheless, the metabolite profile formed after gefitinib injection was comparable to that of PIRF mice with adenoviral deletion of Por (Supplementary Fig. 5). This result similarity indicates that gefitinib has both P450-dependent and -independent drug metabolism.

The biggest and most relevant reduction was observed for O-desmethyl gefitinib (M4, M523595), which is by far the most abundant metabolite in human feces. Rodents produce many different metabolites in addition to M4[40, 41] (Fig. 4b), so we analyzed the M4 metabolite in murine Por-deleted and Por-expressing humanized and non-humanized control mice (Fig. 4c). The highest levels of M4 were detected in murine Por-deficient Hu-PIRF mice, where human hepatocytes preferentially

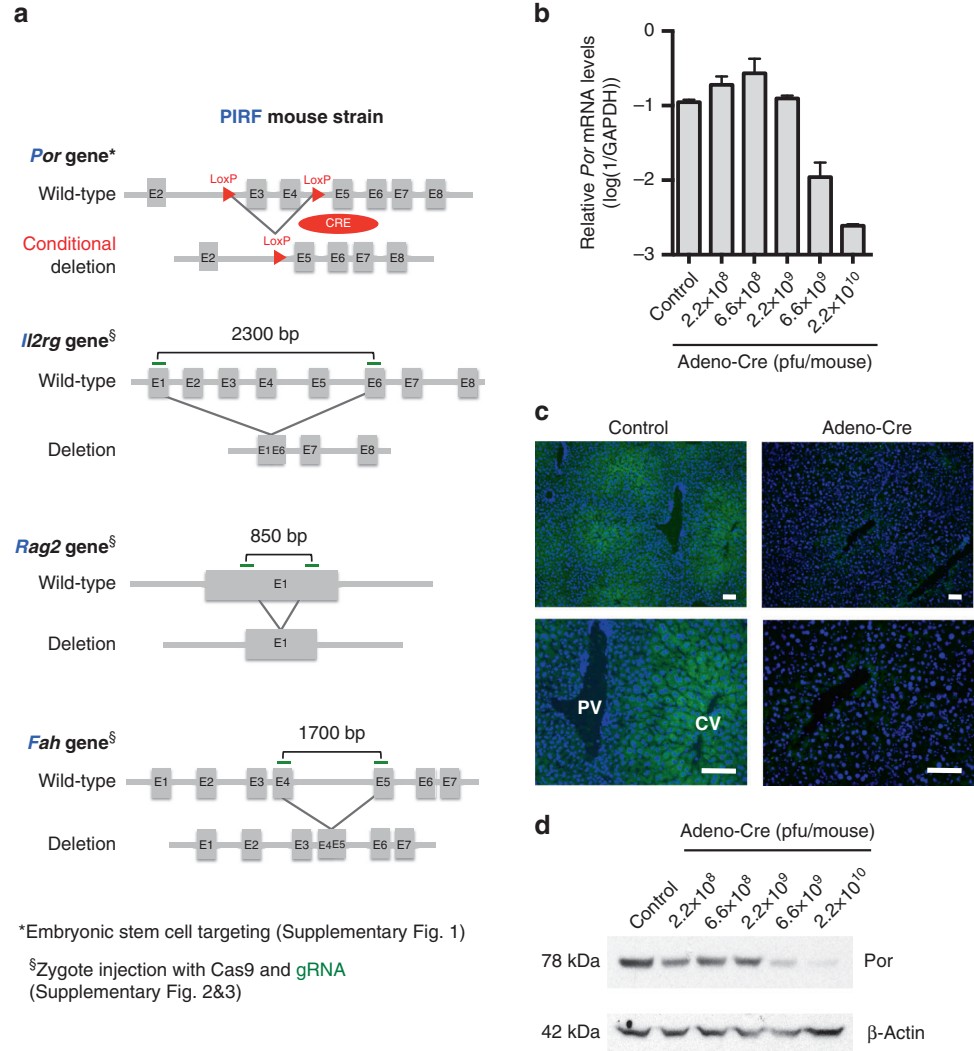

**Fig. 1** Generation of the PIRF strain and deletion of the murine P450 (Por) oxidoreductase. **a** Schematic representation of deleted and transgenic loci in the PIRF strain. **b** qPCR of Por mRNA 7 days after intravenous injection of adenovirus expressing the CRE recombinase (Adeno-Cre). **c** Immunostaining for Por protein demonstrating a gradient across the hepatic acinus with higher pericentral (cv) and lower periportal (pv) expression. Seven days after injection with Adeno-Cre, Por is barely detectable. **d** Western blotting confirms almost complete disappearance of Por protein 7 days after injection. Results are expressed in mean values ± s.e.m. of triplicates ($N = 3$). Scale bar 50 μm. PIRF, $Por^{c/c}/Il2rg^{-/-}/Rag2^{-/-}/Fah^{-/-}$

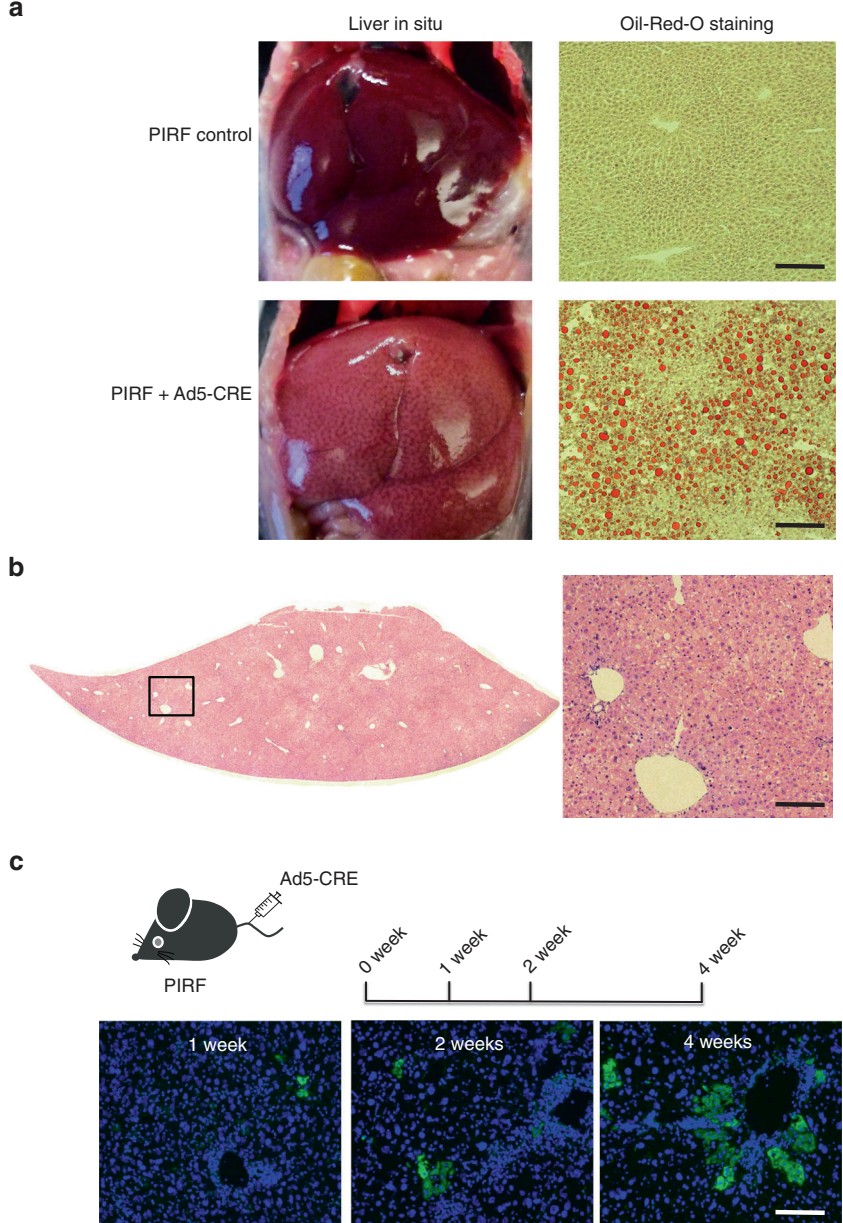

**Fig. 2** Deletion of *Por* with an adenoviral vector expressing CRE recombinase in PIRF mice. **a** 2 weeks after injection of the adenovirus (2.2 × 10¹⁰pfu per mouse) hepatocytes start to accumulate lipids. **b** H&E stain of liver lobe 4 weeks after transduction with adenovirus showing macro and microvesicular steatosis. *Right picture* is a higher magnification of *boxed area* on *left*. **c** Clonal expansion of Por expressing hepatocytes in PIRF mice. Mice were injected intravenously with adenovirus expressing Cre recombinase to delete the floxed *Por* gene. Scale bar 50 μm. PIRF, *Por^{c/c}/Il2rg^{−/−}/Rag2^{−/−}/Fah^{−/−}*

metabolize gefitinib to M4 and the remaining murine hepatocytes are inhibited in their drug metabolism (Fig. 4d). We next sought to measure other human-specific metabolites. The most abundant human metabolite was M28, which could not be detected at all in our non-humanized control mice. Mass spectrometry again showed the highest level of this human-specific metabolite in murine Por-deficient Hu-PIRF mice (Fig. 4e, Supplementary Fig. 7), confirming that these mice showed liver metabolism more similar to humans.

The Por-deficient Hu-PIRF mouse is a novel model system for drug metabolism studies, and we therefore analyzed different body compartments, e.g. the serum (1 h after injection) and the urine for these key gefitinib metabolites. M4 could not be detected in the urine and was massively reduced (23-fold in Hu-PIRF mice) in the serum, while M28 was detectable at lower concentrations in both the urine and the serum of Hu-PIRF

mice (Supplementary Fig. 8a, b). Although present at lower levels in both compartments, M28 mirrored relative abundance observed in feces (Fig. 4d). These findings confirm that gefitinib metabolites are primarily excreted trough the feces[40, 41].

We next sought to confirm human xenobiotic metabolism using liver homogenates of PIRF mice. This time we tested atazanavir, an antiretroviral drug (protease inhibitor) for treatment of human immunodeficiency virus. Our previous studies in human and mouse microsomes demonstrated that atazanavir metabolite M15 is a predominant human metabolite[42]. To determine levels of M15 in humanized PIRF mice, we intravenously injected them with atazanavir and harvested their livers 30 min after injection. M15 levels in *Por*-deleted humanized PIRF mice were 5.4 times greater than those observed in non-deleted mice (Fig. 4f), again indicating that these mice metabolize drugs as humans do.

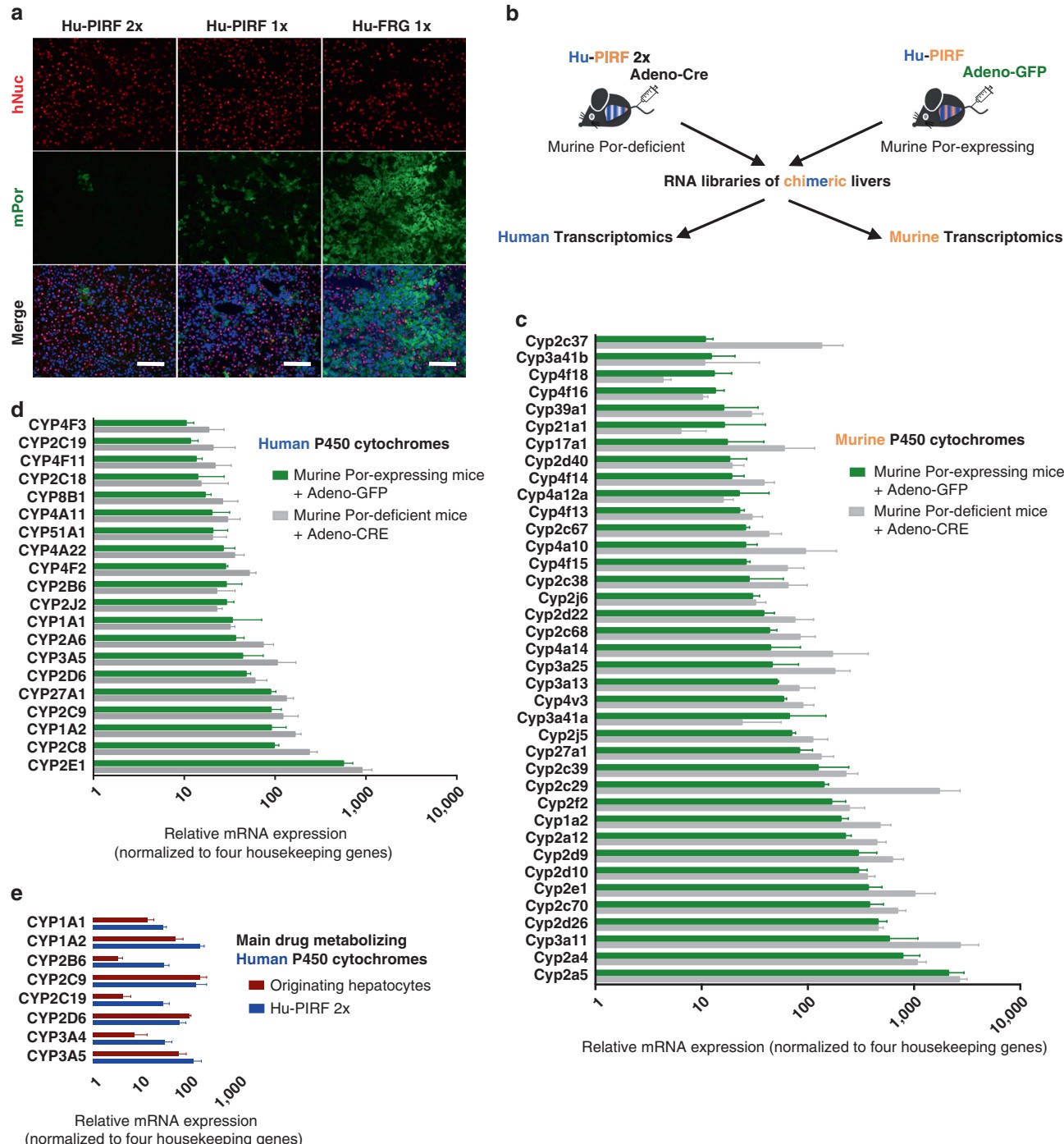

**Fig. 3** Gene expression profiling of humanized PIRF mice following deletion of *Por*. **a** Immunostaining of humanized PIRF and FRG mice for murine Por (mPor) and human nuclei (hNuc) after injection of Adeno-Cre ($2.2 \times 10^{10}$ pfu/mouse) once (1x) or twice (2x). Counterstaining in merged picture using DAPI. **b** Experimental outline for murine and human transcriptomics from chimeric livers with or without *Por* deletion. Chimeric mice ($N = 7$) injected with either Adeno-Cre or Adeno-GFP ($2.2 \times 10^{10}$ pfu/mouse) were euthanized after 1 week and liver tissue from all seven lobes was analyzed. Comparison of murine **c** and human **d** cytochromes originating from the same chimeric livers. **e** Gene expression of the main drug metabolizing human cytochromes in humanized, *Por*-deleted PIRF (Hu-PIRF 2x) mice and originating isogenic human hepatocytes ($N = 2$ donor hepatocytes and 4 isogenic humanized mice). Gene expression has been normalized to four human housekeeping genes and their murine counterparts (*PSMB2*, *PSMB4*, *RAB7A*, and *VPS29*; *Psmb2*, *Psmb4*, *Rab7*, and *Vps29*)[54]. Mean with range are given in **c–e**. FRG, *Fah*[-/-]/*Rag2*[-/-]/*Il2rg*[-/-]; PIRF, *Por*[c/c]/*Il2rg*[-/-]/*Rag2*[-/-]/*Fah*[-/-]

## Discussion

Identification of (reactive) metabolites is crucial for establishing the safety of new drugs, since metabolites drive human drug toxicity[43, 44]. Early and accurate detection of human metabolites is important to mitigate risk and optimize the development of a chemical compound into a drug. Primary assessment of human drug metabolites is usually done using three in vitro systems derived from human liver: liver microsomes, S-9-fractions, or plated human hepatocytes. Each system can predict primary metabolites fairly well, but is limited in predicting more complex multi-step biotransformations[45]. In addition to this major limitation, there is a need for rigorous standardization, since many

experimental parameters such as the incubation time of the drug or protein concentration can bias the analysis[46]. Human hepatocyte-derived in vitro systems are a good first step but cannot replace ADME studies. From an ethical perspective, animal studies are the next step, before moving into clinical trials. The drawback is that experimental animal models have a divergent, usually expanded and poorly characterized set of drug

metabolizing enzymes. With the advent of human liver chimeric mice, a new and attractive option was available to study human drug metabolism in vivo without endangering any humans.

Here we introduce a next generation of humanized mouse model amenable to human drug metabolism with minimal interference from the murine P450 cytochromes. We compared the production of human metabolites for two different drugs

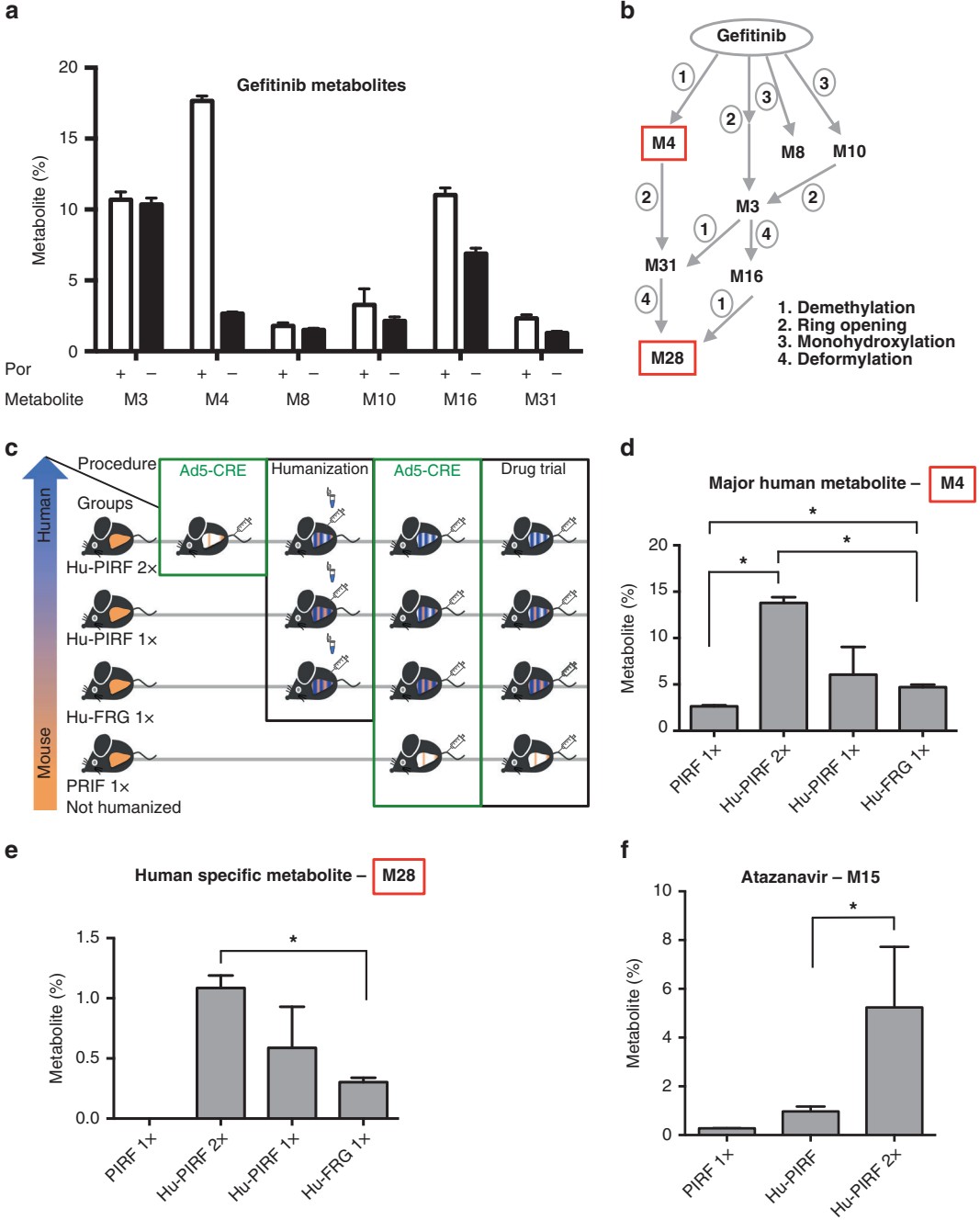

**Fig. 4** Xenobiotic metabolism in humanized PIRF mice. Mass spectrometry analysis of murine feces within 16 h after intravenous injection of gefitinib a cancer therapeutic. **a** Selection of abundant and decreased gefitinib metabolites upon murine P450 oxidoreductase (Por) deletion in non-humanized PIRF mice (N = 3). **b** Gefitinib metabolites and known biotransformation. **c** Experimental setup: not-humanized PIRF mice and Humanized (Hu) FIRF and FRG mice were injected once (1x, after transplantation of human hepatocytes) or twice (2x, before transplantation and after reaching high human chimerism) before doing drug studies. All experimental groups contain three animals, if humanized, with high (>70%) and similar human chimerism (see Methods). Murine Por-deleted, human liver chimeric PIRF (Hu-PIRF 2x) mice show the most abundant human metabolite, M4 **d**, and the human-specific metabolite M28 **e** of gefitinib (N = 3). **f** Mass spectrometry analysis of PIRF liver homogenates 30 min after injection of atazanavir, a retroviral therapeutic. A major human metabolite (M15) is shown. The overall abundance of metabolites from atazanavir or gefitinib was set as 100% in each sample. The data are expressed as mean ± s.e.m. *P < 0.05 using non-parametric Mann–Whitney test. FRG, $Fah^{-/-}/Rag2^{-/-}/Il2rg^{-/-}$; PIRF, $Por^{c/c}/Il2rg^{-/-}/Rag2^{-/-}/Fah^{-/-}$

between humanized PIRF mice and "normal" humanized FRG mice. Analyses revealed higher concentrations of human metabolites in murine feces and liver homogenate in humanized PIRF mice than in FRG mice and demonstrate that these mice have humanized drug metabolism. The PIRF and FRG strains used in this study are in a mixed (C57B and 129 S) genetic background. Aside from potential differences in the background to the two previously published FRG mouse strains[5, 7], our CRISPR/Cas9 generated knock-out strains do not express any transgenes, e.g. the neomycin phosphotransferase, that inactivates a wide range of aminoglycoside antibiotics. This model system should be useful for early detection of reactive metabolites and shape early drug development approaches. Our results are encouraging, and while this is an elegant way to block a large and confounding cluster of drug metabolizing murine enzymes, there is room for improvement. First, adenovirus mediated *Por* deletion occurs primarily in the liver, which is the major but not the only site of cytochrome metabolism. Although a complete *Por* knock-out is embryonic-lethal and might not be compatible with life even after development, knocking out *Por* in a combination of multiple organs like the gut and the liver or the lung and the liver would be desirable. Another consideration is that drug metabolism is not limited to cytochrome metabolism, and additional deletions in other drug-metabolizing enzymes could improve this system. Lastly, it is conceivable that *Por* deletion could be achieved more efficiently. Using transgenic mice expressing Cre recombinase would require yet another crossing step into a quadruple transgenic (PIRF) mouse, however, and an early organ-specific deletion might not generate a robust strain amenable to xenotransplantation.

In summary, we have generated a novel mouse model combining human chimerism with functional deletion of all murine cytochromes by *Por* deletion. Such a murine *Por*-deficient humanization could also be used in combination with other repopulation models such as the transgenic uPA mouse[11, 21]. Our data with two different drugs in two different body compartments demonstrate that studies in humanized PIRF mice efficiently identify human metabolites. Future work will determine how robust this novel system is and demonstrate its limitations. These next-generation human liver chimeric mouse models should promote the development of safer and more efficacious drug therapies.

## Methods

**Generation of the Por-floxed mouse strain**. *Por* knock-out first targeting vector was purchased from the National Institutes of Health Knock-Out Mouse Program (Supplementary Fig. 1). The vector was linearized with the AsisI restriction enzyme, and DNA was electroporated into Jm8A3 mouse ESCs[47] by the Mouse Embryonic Stem Cell Core at Baylor College of Medicine. Integrated clones were selected using neomycin resistance. DNA of ESC clones was digested with NSiI restriction enzyme and screened for site-specific integration by Southern blotting using DIG nonisotopic detection system (Roche Applied Biosciences) following the manufacturer's instructions (full blots in Supplementary Fig. 9). The 500 bp-size 5′ and 3′ probes that bind outside the vector's homology arms were synthesized using the following set of primers:

5′-*Por* Fw2 GGCCTCAGAGAGGACATAGTGCCC-3′
5′-*Por* Rev2 GCCCTCTGGTGTCAGGTCCC-3′
3′-*Por* Fw2 CCTCACGCAGCTTAATGTGGCC-5′
3′-*Por* Rev2 GGAAGTTAAGGACGTGATTACAGGGAGC-5′

Correctly targeted ESCs cells were injected into C57/BL blastocysts by the Genetically Engineered Mouse Core at Baylor College of Medicine. The male chimeras were bred with C57/BL albino females (Taconic) to visualize germline transmission of targeted ESC. To remove the FRT-flanked LacZ and the neomycin cassette and generate a conditional *Por* knock-out strain (*Por^{c/c}*), we crossed the mice with a *Rosa26-FLPe* strain[30]. Genotyping was performed by Transnetyx (Cordova, TN).

**X-Gal staining**. Embryos and fresh liver sections were fixed in 4% PFA for 1 h at 4℃ and washed 2 × 30 min in X-Gal rinse buffer (PBS 1× with 0.02% Igepal and

0.01% deoxycholate) followed by overnight incubation with X-Gal staining solution (PBS 1× with 5 mM K$_3$Fe(CN)$_6$, 5 mM K$_4$Fe(CN)$_6$, 0.02% Igepal, 0.01% deoxycholate, 2 mM MgCl$_2$, 5 mM EGTA, and 1 mg per ml of fresh X-Gal). Samples were post-fixed overnight in 4% PFA at 4℃.

**Generating PIRF (Por^{c/c}/Il2rg^{−/−}/Rag2^{−/−}/Fah^{−/−}) mice**. Six gRNA sequences targeting critical exons of the *Rag2*, *Il2rg*, or *Fah* gene were selected (Fig. 1a, Supplementary Figs 2 and 3) using two different online tools (crispr.mit.edu and COSMID[48]). Complementary oligonucleotides were annealed and ligated into the DR274 vector[49] (Addgene plasmid # 42250) using standard molecular cloning techniques with the restriction enzyme BsaI (NEB) and T4 DNA Ligase (NEB). A T7 bacterial promoter sequence was inserted into the pX330-U6-Chimeric_BB-CBh-hSpCas9 vector[50] (Addgene plasmid # 42230) upstream of the Cas9 transcription start site using standard molecular cloning techniques. DR274 vectors were cut using DraI (NEB) and gelpurified using the Zymoclean Gel DNA Recovery Kit (Zymo, Cat#11-301). In vitro transcription of sgRNA was performed using the MEGAshortscript T7 Transcription Kit (Life Technologies, AM1354) according to manufacturer's instructions. The resulting RNA was purified using the RNA Clean & Concentrator-5 (Zymo, R1015) and eluted in RNAse-free water. Synthesis was verified by polyacrylamide gel electrophoresis. pX330 (with T7 promoter) was digested with NcoI and NotI, and gel purified. Cas9 mRNA was synthesized from the digested pX330-T7 vector using the mMessage mMachine T7 ULTRA Kit (life tech AM1345), according to the manufacturers protocol. Polyadenylation was verified by denaturing agarose gel electrophoresis (1% agarose and 6.6% formaldehyde in MOPS buffer).

Zygotes from *Por^{c/c}* mice were injected with *S. pyogenes* Cas9 mRNA (60 ng/ul) and the six gRNA (15 ng/ul each). All viable zygotes were implanted into three pseudopregnant females. To detect the deleted regions, all 23 pups were genotyped after weaning using the following primers:

*Fah* Fw CTGGGTTGCATACTGGTGGG
*Fah* Rev AAACAGGGTCTTTGCTGCTG
*Fah* Int Fw ACAAAGGTGTGGCAAGGGTT
*Il2* Fw CCACCGGAAGCTACGACAAA
*Il2* Rev GGGGGAATTGGAGGCATTCT
*Il2* Int Rev CTTCTTCCCGTGCTACCCTC
*Rag2* Fw CCTCCCACCTCTTCGTTATCC
*Rag2* Rev AGTCTGAGGGGCTTTTGCTA
*Rag2* Int Fw AGTCTGAGGGGCTTTTGCTA

Further offspring genotyping was performed by Transnetyx (Cordova, TN).

**Humanization of PIRF mice**. Hepatocytes (3 × 10$^6$ per mouse) purchased from Triangle Research Labs/Lonza were transplanted into the murine liver of female PIRF mice by splenic injection, as originally described for mouse hepatocytes[51]. In brief, the abdominal cavity was opened by a midabdominal incision, and 3 × 10$^6$ human hepatocytes in a volume of 100 μl HCM were injected into the spleen. Immediately after transplantation, selection pressure toward transplanted human hepatocytes was applied by withdrawing the drug nitisinone (NTBC) from the drinking water in the following steps: 2 days at 25%, then 2 days at 12%, and eventually 2 days at 6% of the colony maintenance dose (100% = 7.5 mg/l) prior to discontinuing the drug completely[20]. Mice with clinical symptoms (hunched posture, lethargy, weight loss, etc) were put back on 100% nitisinone for a few days before once again being weaned off the drug as described above. In order to determine the extent of human chimerism, we measured human albumin (ELISA, Bethyl laboratories) in the murine blood, having previously shown that human albumin levels correlate with the level of human chimerism assessed by immunostaining of human hepatocytes[20]. Only mice with a human chimerism >70% were used for this study. Where indicated, some PIRF mice were injected intravenously with 100 μl Adenovirus coding CRE recombinase under the CMV promoter (Ad5 CMV-Cre, 2.2 × 10$^{10}$ pfu/ml, provided by the Vector Development Laboratory at Baylor College of Medicine) either 24 h before hepatocyte transplantation and/or when reaching high human chimerism (>70%). Available hepatocyte donor information is given in Supplementary Table 1. All animal experiments were approved by the Baylor College of Medicine Institutional Animal Care and Use Committee. All animals used for humanization (including controls) were female, due to fewer postsurgical complications.

**qPCR**. Total mRNA was isolated from fresh frozen tissue samples using Purelink RNA mini kit (Invitrogen). Two micrograms of total mRNA was reverse-transcribed using the qScript cDNA supermix (Quanta Biosciences) and 20 ng of cDNA was used for the qPCR reactions, performed with Perfecta SYBR Green Fast Mix (Quanta Biosciences) and analyzed on ABI Prism 7900HT Sequence Detection System (Applied Biosciences). The following primers were used for *Por* mRNA amplification of PIRF mouse samples:

*mPor* Fw2 GGCCCCACCTGTCAAAGAGAGCAGC
*mPor* Rev1: CAAACTTGACACCCGTGAGGTCC

For humanized PIRF mouse liver samples, mouse *Por* and human *POR* were amplified using the following set of primers:

mPor Fw1: TCTATGGCTCCCAGACGGGAACC
mPor Rev2: CCAATCATAGAAGTCCTGCGCG
hPOR Fw1: CCAATCATAGAAGTCCTGCGCG
hPOR Rev5: ACCTTGGCCGCATCTATGTCGG

Each sample was normalized to *Gapdh/GADPH* as an internal control gene using the following primers:

mGapdh Fw AGAACATCATCCCTGCATCCA
mGapdh Rev CAGATCCACGACGGACACATT
hGAPDH Fw: CAGAACATCATCCCTGCCTCTAC
hGAPDH Rev: TTGAAGTCAGAGGAGACCACCTG

**RNA-Seq libraries**. Whole-transcriptome RNA sequencing (RNA-Seq) was performed using total RNA extracted from fresh-frozen liver tissue sampled from all seven liver lobes. Total RNA was isolated using the Purelink RNA mini kit (Invitrogen). Libraries were generated from total RNA according to the manufacturer's recommendation using the TrueSeq Stranded mRNA LT kit (Illumina). The libraries were sequenced on a NextSeq 500 sequencer. The average read count per sample was 32 million. RNA-Seq TPM expression values were calculated with RSEM[52] (version 1.2.17) using the read aligner Bowtie2[53] applied to the combined human and mouse NCBI Refseq (3/21/16) transcriptomes. Low-abundance cytochromes (human < 20 TPM and mouse < 20 TPM) were only compared if one of the experimental groups reached >20TPM. Gene expression has been normalized to four human housekeeping genes and their murine counterparts (*PSMB2, PSMB4, RAB7A,* and *VPS29*; *Psmb2, Psmb4, Rab7,* and *Vps29*)[54]. RNA-Seq data is available from European Nucleotide Archive, ENA accession code PRJEB14714.

**Western blot**. Tissue from snap-frozen liver was homogenized in RIPA buffer (Sigma, cat# R0278-50 ml) containing protease inhibitors (Roche, cat# 04693159001). Thirty micrograms of total protein was electrophoresed in a NuPAGE 4–12% Bis Tris Gel (Invitrogen, cat# NP0336BOX) and transferred to a PVDF membrane (Millipore, cat# IPVH00010). The blot was then blocked in 5% milk, followed by primary antibody incubation. Rabbit anti-Por (Abcam cat# ab13513) or mouse anti-β-actin (Sigma cat# A1978) were diluted 1:1,000 and 1:3,000, respectively (full blots in Supplementary Figs 10 and 11). Secondary antibodies were donkey anti-rabbit IgG/HRP and donkey anti-mouse IgG/HRP (Jackson Immunoresearch Labs, cat# 711-035-152 and 711-035-150) used at 1:10,000 and 1:50,000, respectively. The membrane was imaged using Amersham ECL Western Blotting Detection Reagent (General Electric Healthcare Life Sciences, cat# RPN2106).

**Immunohistochemistry**. Ten-micrometer sections from cryopreserved tissue blocks were fixed with 3% PFA for 15 min and incubated overnight at 4 °C with the following primary antibodies: anti-Por (Abcam, cat# ab13513) diluted 1:500, anti-human Nuclei (EMD Millipore, cat# MAB1281) diluted 1:250 in PBS containing 0.2% Triton X-100, and 0.5% BSA. Secondary antibodies (1:1,000 Alexa-fluor conjugated, Molecular Probes) were incubated for 60 min at room temperature in the same buffer. Sections were mounted with Vectashield plus DAPI (Vector Labs).

**Sample preparation for mass spectrometry**. Female mice (6- to 10-month old, humanized or non-humanized) were maintained under a standard 12-h dark/light cycle with water and chow provided ad libitum. The mice were treated (i.v.) with gefitinib (10 mg/kg) and housed separately in metabolic cages for 16 h feces collection. Feces samples were weighed and homogenized in water (100 mg feces in 1,000 μl of H₂O) with the internal standard agomelatine (an antidepressant drug). Subsequently, 300 μl of methanol was added to 100 μl of the resulting mixture, followed by centrifugation at 15,000 g for 20 min. The supernatant was transferred to a new Eppendorf vial for a second centrifugation (15,000 g for 20 min). The final concentration of agomelatine is 2 μM. Each supernatant was transferred to an auto sampler vial for analysis (see below).

For atazanavir metabolism in liver, liver samples were harvested 30 min after the treatment of atazanavir (i.v., 30 mg/kg). Briefly, livers were weighted and homogenized in water/MeOH with the internal standard agomelatine (100 mg liver in 300μl of H₂O/MeOH (v/v 3:1)). Subsequently, 300 μl of methanol was added to 100 μl of the resulting mixture, followed by centrifugation at 15,000 g for 20 min. The supernatant was transferred to a new Eppendorf vial for a second centrifugation (15,000 g for 20 min). The final concentration of agomelatine is 2 μM in samples. Each supernatant was transferred to an auto sampler vial. Five microliters of each prepared sample was injected to a system combining ultra-high performance liquid chromatography (UHPLC) and quadruple time-of-flight mass spectrometry (QTOFMS) for analysis.

**Mass spectrometry (UHPLC–QTOFMS analysis)**. Metabolites from gefitinib and atazanavir were separated using a 1260 Infinity Binary LC System (Agilent Technologies, Santa Clara, CA) equipped with 100 × 2.1 mm (Agilent XDB C18) column. The column temperature was maintained at 40 °C. The flow rate of was 0.3 ml/min, with a gradient ranging from 2 to 98% aqueous acetonitrile containing 0.1% formic acid in a 15-min run. QTOFMS was operated in positive mode with

electrospray ionization. Ultra highly pure nitrogen was applied as the drying gas (12 l/min) and the collision gas. The drying gas temperature was set at 325 °C and nebulizer pressure was kept at 35 psi. The capillary voltages were set at 3.5 kV. During mass spectrometry, real-time mass correction and accurate mass were achieved by continuously measuring standard reference ions at *m/z* 121.0508, 922.0098 in the positive mode. Mass chromatograms and mass spectra were acquired by MassHunter Workstation data Acquisition software (Agilent, Santa Clara, CA) in centroid and profile formats from *m/z* 50 to 1000. The acquisition rate was set as 1.5 spectra per s. The method used in this study has been validated by our previous study of gefitinib metabolism in human liver microsomes[39]. Meanwhile, the quality control samples were performed every 10 samples in the process of the sample running. Due to the authentic compounds of metabolites not available, the metabolite identification was based on their exact mass and MS/MS fragments. The chromatograms and relative abundance of metabolite were performed on Qualitative Analysis software (Agilent, Santa Clara, CA). The relative abundance was evaluated based on integrated peak area of each metabolite.

**Statistics**. Sample sizes for experiments were determined by estimated differences between groups and availability of highly humanized mice. No randomization of animals before allocation to experimental groups nor blinding of experimental groups was done. Statistical analysis was performed using PRISM version 6.0 software (Graph Pad software) using Mann–Whitney test or ANOVA. Statistical significance was assumed with a *P* value < 0.05 (*). Bars in graphs represent mean ± s. e.m. unless noted otherwise. Group size (*N*) represents biological sample size.

**Data availability**. All data generated or analyzed during this study are included in this published article (and its supplementary information files) with the exception of RNA Seq data. The RNA-Seq data is available from European Nucleotide Archive, ENA accession code PRJEB14714.

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

## Acknowledgements

We thank Nathan Nowak, Stephanie Sandor, Isabel Lorenzo, Jason Heaney, Rui Chen and Jeffrey Howard, Robert Kruse and Angie Cohns for technical assistance, V.L. Brandt and C. Gillespie for critical comments on the manuscript, and Hoang Nguyen for sharing reagents. K.D.B. is supported by the National Heart Lung and Blood Institute (NHLBI) grant R01HL134510, the Texas Hepatocellular Carcinoma Consortium (THCCC) (CPRIT #RP150587) and the Diana Helis Henry and Adrienne Helis Malvin Medical Research Foundations. F.P.P. and B.B.C. were supported by T32HL092332. The DLDCC is supported by P30CA125123. CMM core facility of Texas Medical Center Digestive Disease Center (P30-DK56338). NCI Cancer Center Support Grant (P30CA125123) for the Mouse Embryonic Stem Cell Core and the Genetically Engineered Mouse Core at Baylor College of Medicine.

## Author contributions

K.D.B. and M.B.: Designed the experiments. M.B. and B.B.C.: Performed in vivo experiments. M.B.: Did immunostaining and Southern blot. D.Y.: Prepared libraries and run sequencing. X. L.: Did western blotting and PCRs. M.B. and X.L.: Did qPCRs. M.B., F.P.P., and K.D.B.: Designed and prepared CRISPR/Cas9 knock-out approach. X. Li. and F.L.: Performed analysis of drug metabolites. B.Z. and P.S.: Did bioinformatics. All authors read and approved the final manuscript.

## Additional information

**Competing interests:** The authors declare no competing financial interests.

**Change History:** An incorrect version of the Supplementary Information was inadvertently published with this Article where the wrong file was included. The HTML has been updated to include the correct version of the Supplementary Information.

