## [Peer Review File · Nature Communications]

Reviewers' comments:

Reviewer #1 (Remarks to the Author):

This manuscript describes a study on the development of a conditional Por-KO, liver humanized chimeric mouse model and its utility for the assessment of drug metabolism. The mouse model was characterized and processed for RNAseq. The metabolic profiles of two drugs, gefitinib and atazanvir, were also examined. It is concluded that this mouse model would replicate human drug metabolism.

Major Comments

1) There are concerns about the rigorousness of this study due to the lack of description of critical information in Methods. For instance, it is unknown if RNAseq was conducted with single or multiple samples from single/multiple animals per group, and what the sample sizes were in this and many other experiments. The variations in the same group could easily overwrite the changes presented for different groups. In addition, there is no description of the number and sex of mice used for metabolism study. It is unknown if sample preparation and LC-MS analysis is validated. There is even no description of use of authentic and internal standards, etc.

2) The title of this manuscript is misleading and somewhat vague and overstated. It was intended to exhaustively recapitulate the human metabolism of gefitinib and atazanvir in this mouse model. However, the conclusions suggested by the title would need to be demonstrated using comparable human samples, rather than defended by metabolism of two example drugs in selected matrix. In addition, the metabolism study was not using an absolute quantification study but a metabolic profiling approach. Therefore, it would just indicate the metabolic profiles of the two drugs examined.

3) To evaluate the metabolism of gefitinib, the authors use mouse feces as the sample matrix, given that the parent drug is primarily excreted in feces. However, this does not consider possible metabolites that may not be readily eliminated by biliary excretion (e.g., some phase I metabolites that must be further conjugated to enhance elimination). This may not be a major flaw, however, the authors do not sufficiently defend feces as a sample matrix used to pursue further metabolic studies. Nevertheless, the authors should simply investigate metabolites and pharmacokinetics using an alternative sample matrix (e.g., plasma), and compare with human samples/data.

Minor Comments

1) In figure 3e, the authors should not underscore the enhanced mRNA expression of CYP1A2 and 2B6 – this is largely ignored in the results section.

2) The “Humanization of PIRF mice” section: were the human hepatocytes from one donor or several donors? If there were several donors, were the hepatocytes from different donors implanted separately to different mice? The information of the donor(s) should be provided.

3) The result section, “We confirmed expression from the targeted Por locus using the lacZ expression cassette in the embryo and adult liver (Supplementary Fig. 2c)”, “Supplementary Fig. 2c” should be “Supplementary Fig. 1c”, and there is no supplementary Fig. 2c.

Reviewer #2 (Remarks to the Author):

In this study, Barzi and colleagues generated PIRF mice lacking Por, Il2rg, Rag2, and

Fah genes to establish a better mouse model for preclinical drug development. So far the human hepatocyte-transplanted mice have been utilized as the humanized mice for drug development, but they contain P450 cytochromes produced from the remaining murine hepatocytes. The authors improved the humanized characteristics of these mice by conditionally disrupting the *Por* gene, the only one electron donor of all cytochromes.

Overall, the concept is very clear and promising, and there is no doubt about the usefulness of highly humanized mice. However, the evidence of the advantage of PIRF mice over other existing ones such as *Cyp3a*-deficient humanized mice is not sufficient in the current form of the manuscript, failing to show the absolute impact of the study.

Major:

1) The authors performed transcriptomic analysis and quantification of metabolites among humanized or not humanized PIRF and FRG mice. However, there is another improved system reported (reference 22,23). Although the authors touched these references in Introduction, there is no data showing direct comparison against this system. To truly prove the advantageous property of the authors' system, the actual data have to be included.

2) Similar to the above point, the authors indicated in Introduction that *Cyp3a*-KO humanized mice upregulated several other cytochrome clusters. However, near a half of mouse cytochromes are upregulated in the PIRF humanized mice, too (Figure 3C). This reviewer thinks that high quality of this study to satisfy publication criteria of Nature Communications is not guaranteed without some additional data proving relative advantages.

Minor:

3) In Supplementary Table 1, detailed information for Weng et al. 2005 should be provided.

4) The timing of sampling should be described for qPCR, IHC, and WB.

5) Remove "FRG; *Fah*^{-/-}/*Rag2*^{-/-}/*Il2rg*^{-/-}" from the legends to Figures 1 and 2.

6) Add scale bars to Figure 1C.

Reviewer #3 (Remarks to the Author):

The authors created and tried to characterize a new humanized mouse model with murine P450 oxidoreductase deletion. Application of chimeric mice with humanized liver has been well established as a useful *in vivo* platform for predicting human drug disposition or for toxicity test. It is an attractive idea to remove murine P450 oxidoreductase to reduce the P450 activities from mouse. However, the authors didn't clearly demonstrate what has been claimed for.

The results indicated the murine P450 oxidoreductase (*Por*) was not completely knocked out. The "residual signal could be detected by Western blotting even at the highest dose

used (Fig. 1d)". This has also been reflected on Fig 3a, which showed not complete deletion of the Por gene. The most importantly, the authors also compared P450 activities in Por positive and Por deletion non-humanized PIRF mice using gefitinib as a probe drug (Fig 4a). The results didn't show significant gefitinib metabolite reduction except for M4 in Por deletion mice (Fig 4a), suggesting Por may still exhibit relative high level in Por deletion mouse.

In addition, more studies are suggested to demonstrate Por deletion in humanized PIRF mice

(1) mRNA and western blot should be used to test for Por again in humanized PIRF mice, the P450 oxidoreductase difference in mouse and human should be evaluated in certain way as well

(2) The replacement ratio for human P450 gene in chimeric mice should also be evaluated

(3) A similar activity tests as Fig 4d, e, f should be tested in humanized mouse hepatocytes as well as in comparison with those observed the human hepatocytes that was used in this humanization. The author should also provide information about human hepatocytes used in this study, such as age, gender etc.

Point by point response to referees

We thank the reviewers for their helpful feedback. We have addressed their specific comments in multiple additional experiments, resulting in a new Fig. 3, five additional supplementary figures (Supplementary Fig.4-8) and one additional table (Supplementary Table 1). We think these additions significantly improve the paper.

Reviewer #1 (Remarks to the Author):

This manuscript describes a study on the development of a conditional Por-KO, liver humanized chimeric mouse model and its utility for the assessment of drug metabolism. The mouse model was characterized and processed for RNAseq. The metabolic profiles of two drugs, gefitinib and atazanvir, were also examined. It is concluded that this mouse model would replicate human drug metabolism.

Major Comments

1) There are concerns about the rigorousness of this study due to the lack of description of critical information in Methods. For instance, it is unknown if RNAseq was conducted with single or multiple samples from single/multiple animals per group, and what the sample sizes were in this and many other experiments. The variations in the same group could easily overwrite the changes presented for different groups. In addition, there is no description of the number and sex of mice used for metabolism study. It is unknown if sample preparation and LC-MS analysis is validated. There is even no description of use of authentic and internal standards, etc.

Answer: In an attempt to make the approach more appealing to a non-specialized, broader scientific audience, we omitted detailed information, but agree with the reviewer that methods were thus over simplified. We want to assure the reviewer that this was not related to rigorousness of this study, and we have subsequently updated the paper with the following information:

- 1) Information on sex and numbers of mice is given throughout the manuscript and in the methods section.

- 2) We give in this updated version of the manuscript all available information on human hepatocytes and where they were used throughout the study –Supplementary Table 1.
- 3) All LC-MS studies were run with internal standard (agomelatine) added upon sample preparation right from the beginning (see updated Methods section). We also provide additional peak area information on metabolites (Supplementary Fig.5, Supplementary Fig.7 and Supplementary Fig. 8).
- 4) Previous experiments were isogenic duplicates from the same donor. We now addressed the criticism with additional RNAseq experiments from another group of isogenic animals and hepatocytes, increasing the number, but also providing a better control. Instead of using no Adenovirus as a control, we use an Adno-GFP virus. The results are largely the same, but the rigorousness is improved. We would like to emphasize that it is a lengthy (6 months) and difficult process to get high and uniform human chimerism in all mice so that they can be compared to each other. The advantage is that the control group (Adeno-GFP) contains isogenic human hepatocytes and inter-individual variations are not an issue, as with the inbred mouse cytochromes.

We thank the reviewer for these great comments and feel the manuscript is significantly improved.

2) The title of this manuscript is misleading and somewhat vague and overstated. It was intended to exhaustively recapitulate the human metabolism of gefitinib and atazanavir in this mouse model. However, the conclusions suggested by the title would need to be demonstrated using comparable human samples, rather than defended by metabolism of two example drugs in selected matrix. In addition, the metabolism study was not using an absolute quantification study but a metabolic profiling approach. Therefore, it would just indicate the metabolic profiles of the two drugs examined.

Answer: Although we are enthusiastic about our novel mouse model and believe that the first steps are taken to demonstrate human drug metabolism, we now have a more descriptive title “Humanized mice lacking

murine P450 oxidoreductase: a novel model to study human drug metabolism”.

The reviewer raises a very valid point by suggesting a direct comparison to human samples. Unfortunately, these samples can only be collected when performing a clinical study, which will take more than a year. Most importantly, when comparing to human samples, we can not perform the drug study in an isogenic context – our hepatocyte donors are cadaveric (see Supplementary Table 1). Assuming we would have human control samples, they would be very different genetically (many polymorphisms in drug metabolizing enzymes) and we would need to include many humans. Our comparisons are isogenic and every group and control group has the same hepatocytes.

We agree that having absolute numbers in addition to percentage metabolites would be instructive. We now provide several supplementary figures (Supplementary Fig. 5, 7 and 8) with peak area information to complement our data set.

3) To evaluate the metabolism of gefitinib, the authors use mouse feces as the sample matrix, given that the parent drug is primarily excreted in feces. However, this does not consider possible metabolites that may not be readily eliminated by biliary excretion (e.g., some phase I metabolites that must be further conjugated to enhance elimination). This may not be a major flaw, however, the authors do not sufficiently defend feces as a sample matrix used to pursue further metabolic studies. Nevertheless, the authors should simply investigate metabolites and pharmacokinetics using an alternative sample matrix (e.g., plasma), and compare with human samples/data.

Answer: We based our decision to examine feces as the primary compartment on previous drug studies with gefitinib in humans and other animals (ref#40 and #41). Nevertheless, we collected from our liver chimeric animals and control groups urine and serum and analyzed the metabolites as suggested by the reviewer. We found very few metabolites in these compartments and complemented the manuscript with absolute numbers where available (paragraph in results: “other compartment” and Supplementary Fig. 7 and 8).

Minor Comments

1) In figure 3e, the authors should not underscore the enhanced mRNA expression of CYP1A2 and 2B6 – this is largely ignored in the results section.

Answer: We could confirm higher expression in these clusters with additional sequencing (new Fig. 3e), and are drawing the attention of the reader to this observation in the results section of the new manuscript version.

2) The “Humanization of PIRF mice” section: were the human hepatocytes from one donor or several donors? If there were several donors, were the hepatocytes from different donors implanted separately to different mice? The information of the donor(s) should be provided.

Answer: In the current version of the manuscript, we provide all donor information in Supplementary Table 1 and mention that all experiment were done with isogenic control groups, e.g. all control groups had hepatocytes from the same donors to account for inter-individual variation of expression as correctly noticed by the reviewer.

3) The result section, “We confirmed expression from the targeted Por locus using the lacZ expression cassette in the embryo and adult liver (Supplementary Fig. 2c)”, “Supplementary Fig. 2c” should be “Supplementary Fig. 1c”, and there is no supplementary Fig. 2c.

Answer: We corrected our mistake – thank you.

Reviewer #2 (Remarks to the Author):

In this study, Barzi and colleagues generated PIRF mice lacking *Por*, *Il2rg*, *Rag2*, and *Fah* genes to establish a better mouse model for preclinical drug development. So far the human hepatocyte-transplanted mice have been utilized as the humanized mice for drug development, but they contain P450 cytochromes produced from the remaining murine hepatocytes. The authors improved the humanized characteristics of these mice by conditionally disrupting the *Por* gene, the only one electron donor of all cytochromes.

Overall, the concept is very clear and promising, and there is no doubt about the usefulness of highly humanized mice. However, the evidence of the advantage of PIRF mice over other existing ones such as *Cyp3a*-deficient humanized mice is not sufficient in the current form of the manuscript, failing to show the absolute impact of the study.

Major:

1) The authors performed transcriptomic analysis and quantification of metabolites among humanized or not humanized PIRF and FRG mice. However, there is another improved system reported (reference 22,23). Although the authors touched these references in Introduction, there is no data showing direct comparison against this system. To truly prove the advantageous property of the authors' system, the actual data have to be included.

Answer: We decided to use “normal” humanized mice as a control since they are increasingly used by many groups, and widely accepted as an optimized system for human drug metabolism (ref#12-19). The humanized *cyp3a* mouse has been recently published and results are so far only reported by one commercial group (Phoenix Bio Co.) in Japan. In contrast to our model, the *cyp3a* humanized mouse has 71 out of 72 murine cytochromes that are fully functional and, as the authors show, there are compensatory upregulations. In our mouse, we also observed altered expression profiles but none of the murine cytochromes are functional due

to the lack of their electron donor (Por), hence the altered expression profile has no functional consequence in contrast to the cyp3a-deficient mouse. We apologize for not making these differences in humanized control mice clear in the previous version of the manuscript.

Also there are several challenges to overcome in order to compare our strain to the cyp3a humanized mouse:

- 1) This mouse is based on a different repopulation mechanism (transgenic uPA mouse and not FRG mouse).
- 2) The cyp3a deficient mouse is generated by a company (Phoenix Bio Co. Japan), which is unlikely to share their technology with us.
- 3) This experiment is likely to take longer than a year: MTA with a company: 2-6 months, shipment & rederivation 2 months, breeding of colony for repopulation: 3-4 months repopulation 4-6 months.

2) Similar to the above point, the authors indicated in Introduction that Cyp3a-KO humanized mice upregulated several other cytochrome clusters. However, near a half of mouse cytochromes are upregulated in the PIRF humanized mice, too (Figure 3C). This reviewer thinks that high quality of this study to satisfy publication criteria of Nature Communications is not guaranteed without some additional data proving relative advantages.

Answer: We performed additional RNAseq to increase the validity of our cytochrome expression studies (new Fig. 3) and as the reviewer notes correctly, many murine cytochromes are significantly altered. However, altered murine cytochromes do not have a functional consequence in our model (only the human cytochromes are functional) since they don't get the electron from the murine Por. In the cyp3a deficient mouse all 72 cytochromes except the deleted cyp3a are functional. We report the murine cytochromes for completeness of the study, although their expression levels are irrelevant. This is in clear contrast to the cyp3a deficient humanized mouse, where these levels have functional relevance. We apologize for not making this point clear in previous version of the manuscript.

Minor:

3) In Supplementary Table 1, detailed information for Weng et al. 2005

should be provided.

Answer: We apologize for not giving detailed information in the previous version of the manuscript, and have updated the Supplementary Table accordingly.

4) The timing of sampling should be described for qPCR, IHC, and WB.

Answer: We have added this information to the manuscript

5) Remove "FRG; Fah^{-/-}/Rag2^{-/-}/Il2rg^{-/-}" from the legends to Figures 1 and

Answer: Thank you, we have removed this from the legend.

6) Add scale bars to Figure 1C.

Answer: Thank you, we have added scale bars to Figure 1C.

Reviewer #3 (Remarks to the Author):

The authors created and tried to characterize a new humanized mouse model with murine P450 oxidoreductase deletion. Application of chimeric mice with humanized liver has been well established as a useful in vivo platform for predicting human drug disposition or for toxicity test. It is an attractive idea to remove murine P450 oxidoreductase to reduce the P450 activities from mouse. However, the authors didn't clearly demonstrate what has been claimed for.

The results indicated the murine P450 oxidoreductase (Por) was not completely knocked out. The "residual signal could be detected by Western blotting even at the highest dose used (Fig. 1d)". This has also been reflected on Fig 3a, which showed not complete deletion of the Por gene. The most importantly, the authors also compared P450 activities in Por positive and Por deletion non-humanized PIRF mice using gefitinib as a probe drug (Fig 4a). The results didn't show significant gefitinib metabolite reduction except for M4 in Por deletion mice (Fig 4a), suggesting Por may still exhibit relative high level in Por deletion mouse.

Answer: Remaining Por activity is a reasonable explanation for the missing reduction of several metabolites, as suggested by the reviewer. We experimentally addressed this concern by crossing the Por^{c/c} mouse to a transgenic alb-CRE expressing mouse. Indeed the Por deletion was better as shown in immunostaining, Western blotting and qPCR (new Supplementary Fig. 6). Nevertheless, some of the gefitinib metabolites still did not differ from Por-expressing mice (new Supplementary Fig. 5). We conclude that gefitinib has Por-dependent and -independent drug metabolism. Thank you for this thoughtful question, addressing which greatly improved our manuscript.

In addition, more studies are suggested to demonstrate Por deletion in humanized PIRF mice

(1) mRNA and western blot should be used to test for Por again in humanized PIRF mice, the P450 oxidoreductase difference in mouse and human should be evaluated in certain way as well.

Answer: We experimentally addressed this shortcoming and updated the manuscript with qPCR and Western blotting of Por in human liver chimeric mice (new Supplementary Fig. 4).

(2) The replacement ratio for human P450 gene in chimeric mice should also be evaluated

Answer: We addressed this question experimentally and evaluated by qPCR the replacement ratio of murine to human POR upon deletion (new Supplementary Fig. 4a).

(3) A similar activity tests as Fig 4d, e, f should be tested in humanized mouse hepatocytes as well as in comparison with those observed the human hepatocytes that was used in this humanization. The author should also provide information about human hepatocytes used in this study, such as age, gender etc.

Answer: Our studies were done with cadaveric human hepatocytes and all groups have the same human hepatocytes (isogenic). We can not do clinical trials with the hepatocyte donors (gefitinib exposure), but we included now all available information of hepatocyte donors in a new table (new Supplementary Table 1) and also compare human gene expression profiles of primary hepatocytes (Fig. 3e).

REVIEWERS' COMMENTS:

Reviewer #1 (Remarks to the Author):

This manuscript presents efforts for the generation and characterization of a new liver humanized chimeric mouse model with murine Por "deletion". While there are various liver humanized chimeric mouse models as well as Por KO and KD mouse models created for drug metabolism research, there is no liver chimeric models established under murine Por KO/KD background. However, data described in this manuscript do not demonstrate the claimed utilities for this new mouse model. There are specific concerns about efficiency and dynamics of Por KO/KD, possible variability, and the timing for consequent experiments. Studies are also warranted to clearly demonstrate the effects of murine Por KO/KD on altered metabolic profiles of gefitinib and whether there are indeed species differences in drug metabolism between murine Por and human POR in the chimeric mice. Difference in levels of M4 production (demethylation) could be solely attributed to variable CYP2D6 activities.

Reviewer #2 (Remarks to the Author):

The points raised by this reviewer were appropriately revised.

Reviewer #4 (Remarks to the Author):

The new humanized mouse model described here represents indeed an significant improvement over existing human liver chimeric mouse models. At times, however, the authors oversell the model a bit. For example, the data presented in the study do not support the statement that "[humanized PIRF mice] use exclusively human cytochrome metabolism" (abstract). The Por gene was only partially inactivated which is also reflected in the observations that the probe drug (gefitinib) still appeared to be partially metabolized by mouse enzymes. The study does not loose much of its novelty by stating this explicitly (and more accurately) throughout the manuscript, especially in the abstract.

The authors should also state what exactly the genetic background of the PIRF mice is as this can influence human xenogeneic engraftment. Also what is the source of the FRG mice? Are these the same that were used in the 2005 and 2010 studies by Dr. Bissig or are these resulting triple mutant mice from the CRISPR targeting approach? If the former what is the genetic background of the mice and if they are not on a isogenic background (compared to the PIRF mice) have the authors observed any differences in human hepatocyte engraftment, survival etc?

Additional minor point: the ILRg is x-linked and thus if the gender is not specified the null allele should be designated as ILRgNULL and not IL2Rg-/- (only in females) as males are IL2Rg-/y.

Point by point response to referees (final submission)

We thank the reviewers for their helpful feedback. We have addressed their specific comments in the final version of the manuscript.

REVIEWERS' COMMENTS:

Reviewer #1 (Remarks to the Author):

This manuscript presents efforts for the generation and characterization of a new liver humanized chimeric mouse model with murine *Por* “deletion”. While there are various liver humanized chimeric mouse models as well as *Por* KO and KD mouse models created for drug metabolism research, there is no liver chimeric models established under murine *Por* KO/KD background. However, data described in this manuscript do not demonstrate the claimed utilities for this new mouse model. There are specific concerns about efficiency and dynamics of *Por* KO/KD, possible variability, and the timing for consequent experiments. Studies are also warranted to clearly demonstrate the effects of murine *Por* KO/KD on altered metabolic profiles of gefitinib and whether there are indeed species differences in drug metabolism between murine *Por* and human *POR* in the chimeric mice. Difference in levels of M4 production (demethylation) could be solely attributed to variable CYP2D6 activities.

Answer: We have shown reduced *Por* mRNA (qPCR – Fig. 1b) and protein (IHC & Western blotting – Fig. 1c&d, Fig. 3a) upon adenoviral (CRE) deletion. Our data demonstrate >95% KO/KD of *Por* and we therefore respectfully disagree that there are “efficiency concerns”. Moreover, we crossed the *POR*^{c/c} to an *ALB-CRE* strain, and show comparable KO/KD efficiency (Supplementary Fig. 6) and gefitinib metabolites (Supplementary Fig. 5) compared to the adenovirally deleted *POR*^{c/c}.

The differences in M4 levels could indeed be attributed to CYP2D6 polymorphisms, however, we used in all mice the same human hepatocytes, and hence can rule out this possibility.

Reviewer #2 (Remarks to the Author):

The points raised by this reviewer were appropriately revised.

Reviewer #4 (Remarks to the Author):

The new humanized mouse model described here represents indeed an significant improvement over existing human liver chimeric mouse models. At times, however, the authors oversell the model a bit. For example, the data presented in the study do not support the statement that "[humanized PIRF mice] use exclusively human cytochrome metabolism" (abstract). The Por gene was only partially inactivated which is also reflected in the observations that the probe drug (gefitinib) still appeared to be partially metabolized by mouse enzymes. The study does not loose much of its novelty by stating this explicitly (and more accurately) throughout the manuscript, especially in the abstract.

Answer: We have adapted the abstract and discussion according to the suggestion of the reviewer.

The authors should also state what exactly the genetic background of the PIRF mice is as this can influence human xenogeneic engraftment. Also what is the source of the FRG mice? Are these the same that were used in the 2005 and 2010 studies by Dr. Bissig or are these resulting triple mutant mice from the CRISPR targeting approach? If the former what is the genetic background of the mice and if they are not on a isogenic background (compared to the PIRF mice) have the authors observed any differences in human hepatocyte engraftment, survival etc?

Answer: We added this information in the discussion.

Additional minor point: the ILRg is x-linked and thus if the gender is not specified the null allele should be designated as ILRgNULL and not IL2Rg-/- (only in females) as males are IL2Rg-/y.

Answer: We have made according correction in the manuscript.